# Effect of Population Density on Personality of Crayfish (*Procambarus clarkii*)

**DOI:** 10.3390/ani14101486

**Published:** 2024-05-17

**Authors:** Li Su, Leiyu Lu, Mengdi Si, Jingjing Ding, Chunlin Li

**Affiliations:** 1School of Resources and Environmental Engineering, Anhui University, Hefei 230601, China; suli021777@163.com (L.S.); luleiyu1949@163.com (L.L.); mengd199804@163.com (M.S.); 2Jiangsu Academy of Forestry, Nanjing 211153, China; 3Anhui Province Key Laboratory of Wetland Ecosystem Protection and Restoration, Anhui University, Hefei 230601, China

**Keywords:** animal behavior, behavioral syndrome, shyness, exploration, aggression

## Abstract

**Simple Summary:**

Population density is a prevalent environmental factor that influences animal behavior, yet its impact on the formation of animal personality remains largely unexplored. In this study, we reared juvenile crayfish under varying population densities and assessed their personality traits (shyness, exploration, and aggression) upon reaching sexual maturity. Our findings indicated the behavioral repeatability was widespread in crayfish at different densities. Shyness was negatively correlated with exploratory behavior in all crayfish, whereas aggression was positively correlated with exploratory behavior in medium- and high-density female crayfish, indicating the presence of behavioral syndromes within the species. Crayfish raised at medium and high densities were less shy, more exploratory, and more aggressive than low-density individuals. These findings suggest that population density can be an important factor in influencing the personality traits of animals.

**Abstract:**

Personality is widely observed in animals and has important ecological and evolutionary implications. In addition to being heritable, personality traits are also influenced by the environment. Population density commonly affects animal behavior, but the way in which it shapes animal personality remains largely unknown. In this study, we reared juvenile crayfish at different population densities and measured their personality traits (shyness, exploration, and aggression) after reaching sexual maturity. Our results showed repeatability for each behavior in all treatments, except for the shyness of females at medium density. There was a negative correlation between shyness and exploration in each treatment, and aggression and exploration were positively correlated in medium- and high-density females. These indicate the presence of a behavior syndrome. On average, the crayfish raised at higher population densities were less shy, more exploratory, and more aggressive. We found no behavioral differences between the sexes in crayfish. These results suggested that population density may affect the average values of behavioral traits rather than the occurrence of personality traits. Our study highlights the importance of considering population density as a factor influencing personality traits in animals and, therefore, might help us to understand animal personality development.

## 1. Introduction

Animal personality refers to repeatable inter-individual differences and intra-individual consistencies in behavior over time and/or across conditions within populations [1]. It reflects the life history strategies of animals and their broad ecological and evolutionary significance [2], making it crucial for studying speciation [3], spatial population dynamics [4], animal health and management [5], mating choice [6], and competition [7]. Among different individuals in a population, animal behavior varies over time and in different environments; this difference is stable and consistent. Consistency does not mean that individual behavior does not change with age or environmental conditions, rather that the behavior of individuals in a given situation or context is consistently different from that of others in the same group [8]. There is a wealth of evidence that various vertebrates and invertebrates exhibit personality [9], which has attracted increasing theoretical and empirical research attention. Studying and quantifying animal personalities and exploring the factors that shape their behavior can help us to understand the mechanisms underlying behavioral differences between individuals within groups and reveal the implications of these differences in ecology and evolution.

Some studies have connected animal behaviors to genetic factors [10,11]. However, environmental influences can impact personality [12], possibly more than genetic elements. Personality can be significantly influenced by early life and adult experiences [13]. Many external and internal factors influence animal personalities [14,15]. Animals living in different environments (e.g., with early complex environment [16], body type [17], temperature [15], and risk predictability [18]) may have distinct personality types and use various behavioral strategies to adapt to their surroundings [19]. For example, mosquitofish (*Gambusia affinis*) reared at high temperatures are more exploratory than those reared at low temperatures [15]. Fish raised in complex environments are shy, less exploratory, and more social than those raised in open environments [16]. When food resources are scarce, animals exhibit more foraging behavior, take more risks, have higher activity levels, and are more aggressive [20]. Populations with different behavioral types are less likely to be affected by environmental changes. When faced with sudden drastic environmental changes, diverse populations are likely to develop phenotypes to cope with the new environment, thus increasing population sustainability [21,22]. Therefore, more empirical studies are required to determine how environmental differences affect animal personality.

Changes in population density are a common environmental factor affecting animals [23]. Animals may exhibit a variety of behavioral responses to environments with different population densities. For example, high densities restrict activity, feeding, and rest in chickens, easily causing pecking and fighting, and affecting the production of laying hens [24]. Furthermore, studies show that high densities intensify competition between aquatic animals for key resources and lead to severe fighting [25,26]. Nile tilapia raised at high densities exhibit higher levels of chronic stress and aggression than those reared at low densities [27]. In addition, at high densities, fish that use reactive coping methods take longer to display exploring behavior [28,29]. Conversely, individuals from low-density populations may be hesitant to explore, which can limit their access to food and impair their growth [30]. Overall, although population density has an important effect on animal behavior, there is a relative lack of research on how it affects animal personalities. However, the specific mechanism of population density affecting animal personality has not yet been fully explored. Therefore, in-depth investigation into the link between population density and animal personality is crucial for more fully understanding animal adaptability and behavioral responses to different environments. Such research is expected to provide a more comprehensive understanding of related issues in ecology and evolution.

With the rapid growth of the aquaculture industry [31], population density has become an important topic [32,33]; concerns around sustainability [34], environmental impact [35], and animal welfare [36] have increased. These concerns mainly focus on intense interactions between animals, water quality, and stocking density. Increasing density can enhance water space use and breeding productivity. However, high stocking density is a chronic stressor; therefore, long-term exposure to high stocking density may induce stress behavior in animals [22,37]; for example, it increases competition for resources among crustaceans [38] (such as living space, shelter, and food), as well as increasing cannibalism [39] and crowding stress [40] (such as low appetite, poor physiological status, and weak resistance to disease). These factors may lead to decreased survival, size, growth rate, immunity, and yield [33,41]. However, the extent to which population density affects animal behavior and harms welfare is unclear. Therefore, behavior can be used as an indicator of animal welfare [36]. For example, low densities may increase the aggressive behavior of African catfish (*Clarias gariepinus*), resulting in increased skin damage [42,43]. Therefore, studying the impact of population density on animal personality will help to develop improved farming strategies to ensure optimal health, welfare, and production performance of aquaculture animals, thereby promoting the sustainability of the aquaculture industry. Furthermore, investigating the effects of population density on animal personality can provide valuable information on ecology and evolution, helping us to better understand the evolutionary mechanisms of animal behavior and personality.

In this study, we investigated how population density influences crayfish traits. Juvenile crayfish were reared at different population densities, and their personality traits, including shyness, exploration, and aggression, were measured at sexual maturity. The effects of population density and sex on these behaviors were investigated by quantifying the repeatability and correlation of the three behavioral traits under different population densities. Previous studies found that crayfish behavior tends to be stable over time and in a given environment [44,45], and adaptive strategies for different densities of crayfish may lead to changes in the associations between behaviors [46]. We hypothesized that crayfish behavior is repeatable at different population densities and that there is a behavioral syndrome during density treatments. High crayfish stocking densities generally increase competition for resources, which may result in changes in behavioral traits [38]. We expected high-density crayfish to be less shy, more exploratory, and more aggressive than medium- and low-density crayfish. And males were more aggressive despite being reared in different densities. This study expands our understanding of the effect of population density on animal personality and provides a new research perspective on the field of ecology and evolution.

## 2. Materials and Methods

### 2.1. Study Animals and Rearing Conditions

We focused on the aquatic crustacean crayfish, which is commonly used as a model animal in behavioral studies because of its extensive distribution [47]. It is a gonochoristic species, and we distinguished between males and females by observing the location of their reproductive pores, located at the base of the fifth pair of abdominal feet in males and at the base of the third pair of abdominal feet in females. After both sexes reached sexual maturity, the first and second gastropods of male crayfish develop into white calcareous tubular arthropods, whereas the first gastropods of female crayfish degenerate and the second gastropods develop into pinnate arthropods. Crayfish are omnivores with flexible foraging strategies. They have a high reproductive output and short development time. If food is sufficient and there is good water quality, temperature, and other suitable environmental conditions, crayfish can take 1–2 months to reach sexual maturity.

The juvenile crayfish tested in this study were procured from Hefei Huanonghui Market (Co, Hefei, China) and transferred to the laboratory of Anhui University (117.18° E, 31.77° N) for indoor cultivation. We first placed the juvenile crayfish into two previously prepared 200 L tanks with oxygenated water for acclimation and then transferred them into 90 rearing tanks (18.5 × 27 × 13.5 cm) after 2–3 days of adaptation. The rearing tanks were filled with 8–10 cm oxygenated water maintained at 25 ± 1 °C, with >5.0 mg/L dissolved oxygen and pH 7.0–8.5. The water was changed every 2–3 days. We paved the bottom of the tank with aquatic mud and added waterweed (*Elodea canadensis*) and crawling frames (to simulate crayfish crawling on land) to enrich the environment. Juvenile crayfish were fed fresh vegetable leaves, soybean meal, corn, and a small amount of yellow meal, while adult crayfish were fed commercial lobster expanded compound feed (Hongda Feed, Yangzhou Co., Ltd.; Yangzhou, China; crude protein ≥ 32%, crude fiber ≤ 8%, crude fat ≥ 4%, crude ash ≤ 15%, lysine ≥ 1.5%, total phosphorus ≥ 1%, calcium 0.5–3.0%, sodium chloride 0.5–2.5%). To ensure a standardized bait supply for crayfish, we fed approximately 5–8 g of feed per individual per day, provided twice daily. Crayfish were exposed to a natural photoperiod (approximately 14:10 L:D).

We divided 90 rearing tanks into equal numbers of groups of different densities: low, medium, and high density with four, eight, and ten crayfish per tank, respectively. Each tank contained equal proportions of male and female crayfish (Figure 1). After three months of rearing, most crayfish reached sexual maturity. To facilitate the recording of the experimental sequence of crayfish, healthy, sexually mature crayfish were randomly selected from tanks of different densities and we marked their backs; marking was completed two days before the start of the experiment. The experimental crayfish were fasted for 12 h before behavioral experiments. At the end of each experiment, the evaluated individuals were returned to their original rearing tanks.

### 2.2. Personality Assessments

#### 2.2.1. Assessment 1: Shyness and Exploration

We conducted tests to measure crayfish shyness and exploration in an opaque (to prevent the effect of specular reflection on the crayfish’s exploratory behavior [48]) plastic tank (60 × 37.5 × 15 cm). Before each experiment, the tank was filled with 8 cm of oxygenated tap water, which was changed after each test to eliminate any potential effects of chemical signals left by previous subjects on subsequent individuals. A white opaque square shelter (15 × 15 cm) was affixed to one end of the tank and a movable trapdoor (15 × 15 cm) attached to a fishing line was installed to allow the experimenter to remotely and gently open the shelter trapdoor to allow the subject to crawl out. The subjects were shielded using an opaque curtain to prevent disruption from external factors. The exploratory behavior of crayfish in a new environment may be influenced by the type of obstacles encountered [49]. So, in order to minimize the impact of adaptation/habit on the outcome of behavior [50], we used new objects for each individual in subsequent tests, including porcelain pieces, small black boxes, and simulated leaves. A camera (Sony HDR-CX510, 55× extended zoom, Sony Corporation, Tokyo, Japan) was mounted above the experimental tank to record the crayfishes’ behavior. Testing was conducted in a laboratory with consistent lighting, a constant temperature (25 °C), and no interference.

Personality research should not only be based on behavioral data at a given time and in a certain situation but should also examine the repeatability of the same behavior in the same individual at different times to analyze and evaluate individual behaviors more accurately [51]. Therefore, we repeated the experiment three times, approximately 7 days apart. In the first experiment, tagged crayfish were selected from a low-density rearing tank. After performing the first experiment for all low-density tanks, subjects were chosen from the medium- then high-density tanks. Individuals repeated the experiment in the order low-, medium-, and then high-density.

In each experiment, a randomly selected crayfish was gently placed in a fixed, enclosed shelter in an opaque plastic tank (60 × 37.5 × 15 cm), and the camera was turned on to record its behavior (Figure 2a). After a 5 min acclimation period, the observer gently pulled a fishing line at a distance to slowly open the trapdoor, which was kept open until the end of the experiment. The timer was started and the video recording continued for 20 min. All of the crayfish tested emerged from the shelter within 20 min. Shyness was measured as the latency to emerge from the initial shelter, which is a widely used measure of shyness; bold subjects emerge more quickly from shelters [11,17]. We considered that the crayfish had emerged from the shelter when their entire body passed over the trapdoor. The camera continued to record for 10 min after the subjects left the shelter to measure their exploratory behavior. We extracted 600 images (one frame per second) from the 10 min motion video and used ImageJ (v.220706, https://imagej.nih.gov/ij/, accessed on 8 October 2022) to track the head position of the crayfish in each frame to describe its motion path [15,16,52]. The total path length was used to quantify the exploration length of each subject. As some studies accept that exploration measurements can include multiple ranges simultaneously, animals should be active during exploration [53]. This test was considered an exploratory behavior because it was conducted in an unfamiliar, novel environment, where the distance covered was used as a measure of exploration [8,12].

#### 2.2.2. Assessment 2: Aggression

We simulated aggressive intraspecific behavior and determined the strength of crayfish aggression by recording and calculating aggression scores. We introduced a crayfish simulation model and used a cylindrical opaque plastic container as the experimental setup for our aggression study, which was lined with a nylon filter cloth to prevent the crayfish from climbing or sliding (Figure 2b). Experimental crayfish were placed in an opaque cylindrical plastic bucket and allowed to acclimate to the environment for 5 min. The researcher then conducted the crayfish aggression experiment by controlling the level of aggression in the model. We classified this aggression into three classes: (i) non-contact aggression, where the model was rapidly placed in front of the crayfish’s head to trigger an immediate defensive response; (ii) contact aggression, where the model’s pincers were used to touch the upper body of the crayfish, replicating an intraspecific struggle; and (iii) clamp aggression, where the experimenter used tweezers to artificially hold the back of the crayfish for three seconds to elicit a response. The three levels of aggression were gradually increased to maximize the likelihood of aggressive behavior from the crayfish (Table 1). Each crayfish was subjected to 10 experiments at each aggression level, resulting in a total of 175 points used to determine the crayfish aggression score. Crayfish were not fed for 12 h before the experiments to prevent potential effects of feeding on aggressive behavior.

Each crayfish was tested for aggression two days after the shyness and exploration behavior experiments. This order of densities tested was the same as that used in the shyness and exploration experiments. The experimenters tested three levels of aggression experiments, each comprising 10 repetitions of each level of aggression. The inter-trial interval was approximately 5 s for the same level of aggression and 10 s for different aggression levels. We repeated the aggression experiments three times, with 7 days between each repeated experiment.

### 2.3. Statistical Analyses

Repeatability measures the consistency of an individual’s behavior [54]. We used the *rpt* function in the *rptR* package [53] to calculate the repeatability of shyness, exploration, and aggression in each population density experiment, and block ID and individual ID were used as random effects. To estimate the confidence interval, the number of parametric bootstrap iterations was controlled by setting the *nboot* argument to 1000 bootstraps. *p*-values based on likelihood ratio tests were used to determine the significance of behavioral repeatability. Behavioral correlation tests were conducted by averaging the three behavioral traits. We used the *corr.test* function in the *psych* package [55] to perform the Spearman correlation test to accurately assess the correlation between the three behaviors.

We used the Shapiro–Wilk test to check whether the data were normally distributed. A generalized linear mixing model (GLMM) was then fitted to examine the effects of population density and sex on the three behavioral traits. Its interaction term was included in the initial model, but was eventually removed because the interaction effect was not significant. After fitting the model, the *glmmPQL* function in the *MASS* package [56] was used for difference testing. The *emmeans* package was used for post hoc comparisons of different population density treatments. All statistical analyses were conducted using R 4.1.0 [57], and the significance level was set at *p* < 0.05.

## 3. Results

### 3.1. Behavioral Repeatability and Correlation

We quantified the behavioral traits of healthy, sexually mature crayfish selected from rearing tanks with different population densities. There were a total of 166 individuals, of which 74 were females and 92 were males. In the different densities, all behaviors were repeatable, except for shyness in medium-density females (Table 2). We also found that shyness was negatively correlated with exploratory behavior in all density treatments (Table 3 and Figure 3). In female crayfish reared at medium and high densities, aggressive behavior was positively correlated with exploratory behavior (Table 3 and Figure 3).

### 3.2. Effects of Factors on Behavioral Traits

There were differences in shyness, exploration, and aggression among crayfish treated with different population densities (Table 4). But we did not find significant differences in behavioral traits between the sexes (Table 4). On average, the crayfish growing in low-density environments were more shy and less willing to actively explore than crayfish growing in medium- and high-density environments. And the higher the population density, the more aggressive the crayfish (Table 4 and Table 5, Figure 4).

## 4. Discussion

Our study provides evidence for personality traits in crayfish and reveals that population density can shape the personality traits of adult crayfish (Table 2, Figure 3 and Figure 4). Animal personality affects aquatic animals’ life processes and outcomes (e.g., behavior, life history, growth, survival, and reproduction) [58,59]. Individual personality traits often determine differences in behavior patterns, habitat use, and dietary preferences [60] and have important ecological and evolutionary implications. Therefore, research on animal personality can help us to understand animal behavior’s diversity, adaptability, and interaction with the environment and it has practical application value for ecosystem protection and management.

Behavioral repeatability pertains to the consistency of an individual’s behavior over time and in various contexts, a phenomenon that is widely observed in animals [12]. Previous empirical research has indicated that over 35% of phenotypic behavioral variations within animal populations can be attributed to differences in individual behaviors [12]. These differences may have significant implications for the evolutionary trajectory of animal groups and the outcomes of natural selection [61]. We found exploration, aggression, and most shyness behaviors were repeatable in all treatments (Table 2). Typically, crayfish consistently exhibit bold traits in environments [45]. For invasive species with social hierarchies, such as crayfish, those that consistently exhibit aggression, boldness, and exploration are more inclined to take risks in the pursuit of food and are better equipped to engage in competitive dominance interactions [62]. These repeatable behaviors in crayfish can reduce the cost of changing behavioral strategies in different population density environments [63], and contribute to the expansion of their territories across different density gradients, ensuring improved resource access and higher reproductive success. Our study also showed that the shy behavior of female crayfish in only medium-density environment did not show repeatability (Table 2), which may be due to the moderate resource supply in the medium-density environment. In relatively stable environments, bold individuals may be more advantageous, while shy individuals are better at flexibly adjusting their behavior strategies according to environmental changes [64]. The presence of these stable individual behavioral differences may contribute to crayfish exhibiting a wider range of niche characteristics and environmental adaptations than local species [65], thus enhancing their ecological versatility.

In all density treatments, shyness was negatively correlated with exploratory behavior (Table 3 and Figure 3), while aggression was positively correlated with exploratory behavior in medium- and high-density female crayfish (Table 3 and Figure 3). Behavioral syndromes refer to a set of interconnected behaviors that reflect the consistency of individual behaviors in multiple situations [12]. When crayfish behaviors are related, they may not evolve independently but as a whole [12,66]. The presence of behavioral correlations can create trade-offs that limit a crayfish’s ability to cope with limited environmental factors, and thus affect individual fitness [54]. It is important to note that variation in behavioral syndromes in populations is not the result of a random evolutionary process, but rather is driven by the adaptive evolution of behavior, which tends to select the best combination of behavioral traits. Our results suggested that bold crayfish were more willing to explore to obtain sufficient resources [67]. And crayfish at the higher density extend their exploration range to the maximum edge by increasing aggression, which may help to protect their living space and obtain high foraging rates [68], resulting in a state-dependent safety mechanism [69]. The presence of this behavioral syndrome may help crayfish to successfully utilize low-productivity habitats and establish populations more quickly [46]. The results of this study provide further evidence for the existence of crayfish behavioral syndromes and help to explain the maintenance of individual differences in behavioral types [12]. The study of behavioral syndromes may provide important insights into the ecological and evolutionary significance of related behaviors as well as their underlying mechanisms.

As expected, the medium- and high-density crayfish were, on average, less shy, more exploratory, and more aggressive than the low-density crayfish (Table 4 and Table 5, Figure 4). To some extent, animal personality determines their range of performance, which means that they may not perform at their best in all situations. Some individuals may exhibit bolder behavioral traits, while others make cautious adjustments in response to changes in the environment. As population density increases, crayfish may compete for social status, priority, food ownership, and space, leading to increased aggression [70]. As mutual contact and fighting between individuals increases, most of their energy is expended in competition for hiding places and resources [71,72], which triggers more stress. Stress can act as an abiotic stressor capable of triggering the release of metabolites and inducing behavioral responses in homogenous and heterogeneous individuals. However, it may have negative effects on health-related behaviors in animals. This can also be interpreted as behavioral costs [73]. Resources (i.e., food and habitat) are more limited at high than low density, so crayfish in high-density populations may be more willing to take risks and actions to actively seek out novel environments [74,75]. This may increase their access to resources and improve both their survival and reproductive success. Conversely, individuals who lack exploratory and aggressive behaviors may have difficulty accessing food and growing normally [30]. Therefore, behavioral differences in crayfish at different densities may determine their adaptive strategies and behavioral patterns. The results of this study demonstrate the important role of population density in shaping animal personality.

We did not find an effect of sex on crayfish personalities (Table 4). Some studies have shown that sex may affect behavioral traits in animals [76], and hormones have been found in crustaceans; however, their relationship with gonads and humoral substances has not yet been determined [77]. During mating, male crayfish are always dominant and show boldness and aggression, which may be related to their role in reproductive competition, whereas females exhibit a passive response, either accepting or rejecting [78]. Male crayfish tend to occupy larger territories, whereas females tend to nest in smaller areas [79]. In general, both female and male crayfish are aggressive, and female crayfish often compete with each other or with males [80]. However, female crayfish do not attack each other as frequently as males. Due to the potential behavioral differences between the sexes, future research can explore how sex affects personality traits differently in crayfish.

## 5. Conclusions

Our study provided evidence for personality and behavioral syndromes in crayfish. We found repeatability for each behavior in all treatments, except for the shyness of females at medium density. Shyness was negatively correlated with exploratory behavior in all crayfish, whereas aggression was positively correlated with exploratory behavior in medium- and high-density female crayfish, indicating the presence of behavioral syndromes. Crayfish raised at medium and high densities were less shy, more exploratory, and more aggressive than low-density individuals. This may be due to the increase in population density and limited resources; crayfish may show more competitive behaviors in order to survive and reproduce, such as aggression and fighting, to compete for limited resources, resulting in significant differences in behavior for different densities of crayfish. Perhaps due to the limited experimental conditions, we did not observe an effect of sex on these behavioral traits at different densities, and follow-up studies can further explore the effect of population density on the personality of different sexes of crayfish. These findings suggest that population density is a key factor in the formation of crayfish personality. This contributes to our understanding of crayfish personality development and plays an important role in developing research on influencing animal personality and altering average behavioral traits.

## Figures and Tables

**Figure 1 animals-14-01486-f001:**
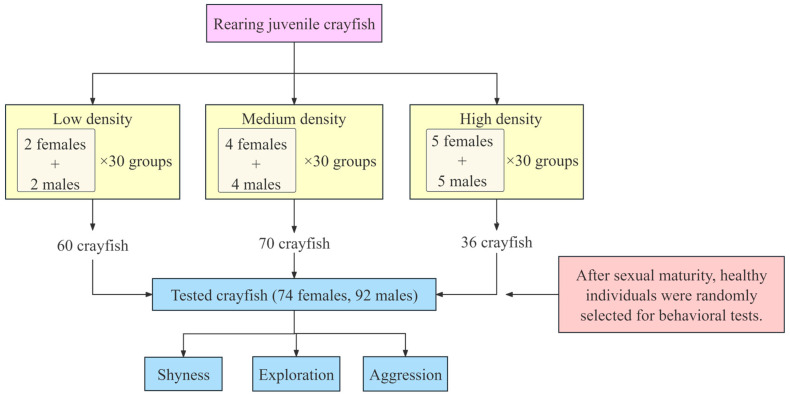
Flow chart of the experiment.

**Figure 2 animals-14-01486-f002:**
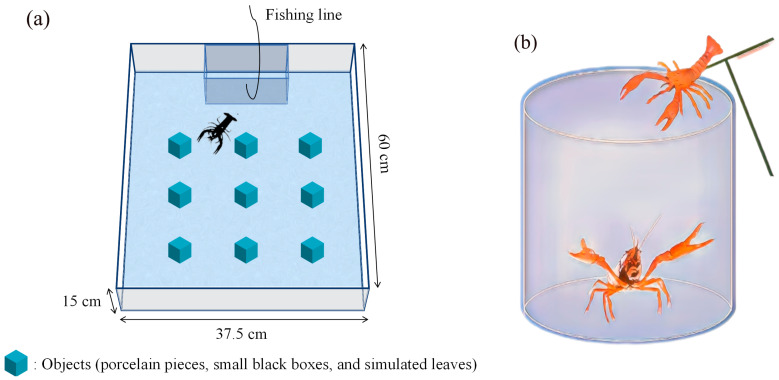
Schematic diagram of personality assessments: (**a**) photograms of the tank for measuring shyness and exploratory experiment (top view); (**b**) photograms of aggression assessments.

**Figure 3 animals-14-01486-f003:**
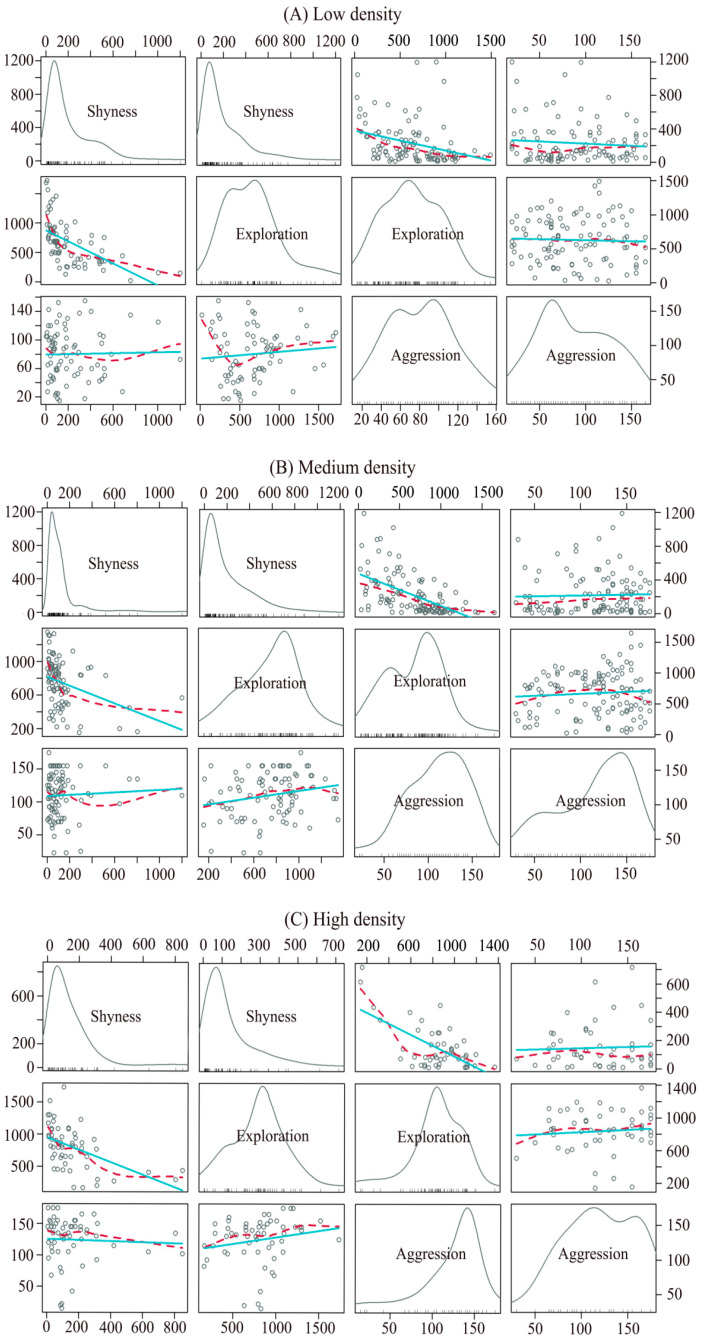
Behavioral correlations of crayfish reared at different densities (lower left: females; top right: males). Each diagonal line in the figure is a kernel density plot for each variable, which is used to better visualize the distribution of the data. On the non-diagonal line are scatter plots and linear fit plots between the different personalities of the crayfish, which represent the degree of correlation between the different behaviors.

**Figure 4 animals-14-01486-f004:**
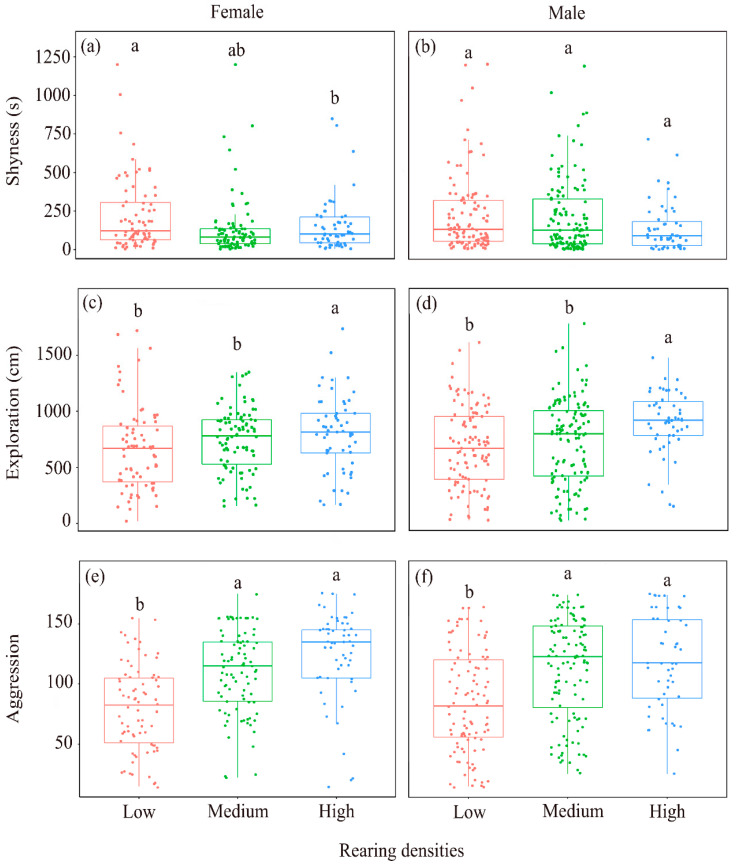
The average behavior level of crayfish reared at different population densities, which was used to compare the distribution characteristics of crayfish personality data in different rearing density environments. (**a**): The differences of crayfish shyness in females; (**b**): The differences of crayfish shyness in males; (**c**): The differences of crayfish exploration in females; (**d**): The differences of crayfish exploration in males; (**e**): The differences of crayfish aggression in females; (**f**): The differences of crayfish aggression in males. Different letters indicate statistical difference between behaviors, while the same letters indicate no statistical difference between behaviors (*p* < 0.05).

**Table 1 animals-14-01486-t001:** Behaviors and scores of crayfish during aggression experiments.

Behaviors	Aggression Levels	Scores
Raising claws	Non-contact aggression	10
	Contact aggression	5
	Clamp aggression	2.5
No raising claws	Non-contact aggression	0
	Contact aggression	0
	Clamp aggression	0

**Table 2 animals-14-01486-t002:** Repeatability of behavioral traits of crayfish in different population densities.

Sex	Repeatability	Shyness			Exploration			Aggression		
		Low	Medium	High	Low	Medium	High	Low	Medium	High
Female	R	0.30	0.04	0.43	0.34	0.19	0.74	0.89	0.84	0.48
	Standard error	0.14	0.08	0.16	0.17	0.12	0.15	0.04	0.11	0.18
	95% confidence intervals	0, 0.51	0, 0.27	0.06, 0.66	0, 0.62	0, 0.42	0.29, 0.85	0.80, 0.94	0.49, 0.91	0.13, 0.80
	*p*-value	**0.008**	0.355	**0.001**	**0.010**	**0.047**	**<0.001**	**<0.001**	**<0.001**	**<0.001**
Male	R	0.21	0.19	0.63	0.53	0.50	0.71	0.91	0.89	0.52
	Standard error	0.12	0.98	0.16	0.14	0.12	0.10	0.13	0.08	0.19
	95% confidence intervals	0, 0.45	0, 0.376	0.18, 0.78	0.21, 0.73	0.24, 0.69	0.43, 0.85	0.48, 0.94	0.61, 0.93	0.14, 0.82
	*p*-value	**0.045**	**0.011**	**<0.001**	**<0.001**	**<0.001**	**<0.001**	**<0.001**	**<0.001**	**<0.001**

Significantly repeatable behaviors are displayed in bold.

**Table 3 animals-14-01486-t003:** Behavioral correlations of crayfish treated with different population densities.

Sex	Population Density	Behavioral Correlation	Correlation Matrix	*p*-Value
Female	Low	**Shyness–Exploration**	**−0.708**	**<0.001 *****
		Shyness–Aggression	−0.040	0.731
		Exploration–Aggression	0.077	0.512
	Medium	**Shyness–Exploration**	**−0.432**	**<0.001 *****
		Shyness–Aggression	0.035	0.743
		**Exploration–Aggression**	**0.201**	**0.057 ***
	High	**Shyness–Exploration**	**−0.561**	**<0.001 *****
		Shyness–Aggression	−0.090	0.507
		**Exploration–Aggression**	**0.270**	**0.042 ****
Male	Low	**Shyness–Exploration**	**−0.418**	**<0.001 *****
		Shyness–Aggression	0.047	0.634
		Exploration–Aggression	−0.049	0.619
	Medium	**Shyness–Exploration**	**−0.626**	**<0.001 *****
		Shyness–Aggression	0.056	0.543
		Exploration–Aggression	0.007	0.943
	High	**Shyness–Exploration**	**−0.337**	**0.016 ****
		Shyness–Aggression	−0.044	0.760
		Exploration–Aggression	0.090	0.531

Significant correlations were displayed in bold. (* *p* < 0.1, ** *p* < 0.05, *** *p* < 0.001.)

**Table 4 animals-14-01486-t004:** The effects of population density and sex on crayfish personality.

Personality	Explanatory Variables	Estimate	SE	*t* Value	*p* Value
Shyness	Rearing density (low density)	68.70	26.55	2.59	**0.01 ****
	Rearing density (medium density)	24.70	25.81	0.96	0.34
	Sex	37.23	19.67	1.89	0.06
Exploration	Rearing density (low density)	−167.94	40.22	−4.18	**<0.001 *****
	Rearing density (medium density)	−110.01	39.11	−2.81	0.01 **
	Sex	−38.05	29.80	−1.28	0.20
Aggression	Rearing density (low density)	−37.91	4.66	−8.13	**<0.001 *****
	Rearing density (medium density)	−9.70	4.53	−2.14	**0.03 ****
	Sex	3.97	3.46	1.15	0.25

Significant differences were displayed in bold. (** *p* < 0.05, *** *p* < 0.001.)

**Table 5 animals-14-01486-t005:** Sampling information and average behavioral traits of crayfish reared at different population densities (±standard error, SE).

Sex	Population Density	Sample Size	Shyness (s)	Exploration (cm)	Aggression
Female	Low	25	220.3 (±26.9)	669.1 (±43.5)	80.2 (±4.1)
	Medium	30	133.1 (±19.5)	746.3 (±29.7)	110.3 (±3.6)
	High	19	156.0 (±23.0)	801.9 (±44.5)	124.2 (±4.9)
Male	Low	35	229.5 (±24.5)	629.0 (±32.5)	88.2 (±4.0)
	Medium	40	217.3 (±21.5)	672.4 (±32.0)	115.0 (±3.7)
	High	17	149.3 (±22.0)	835.6 (±36.0)	120.2 (±5.4)

## Data Availability

The datasets generated during and/or analyzed during the current study are available from the corresponding author on reasonable request.

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
