# Peer review of "Effect of Population Density on Personality of Crayfish (Procambarus clarkii)"

_animals, 2024, doi:10.3390/ani14101486_

Round 1
Reviewer 1 Report
Comments and Suggestions for Authors
Personality traits (shyness, exploration, and aggression) of juvenile crayfish under varying population densities upon reaching sexual maturity assessed will support the production effenciency. Many factors actually influence the personality traits, which include the population test, body size, gender and so on. However, you have not given the suggestive density for the production. In addition,due to the net factor-sex,, interactive effect between sex and density should be estimated. What is important factor body size should be involved in this test.
Comments on the Quality of English LanguageThis paper quality is not enough for publication in Animals. The research design is simple and unadquate innovation so that it is rejected. How to select the best pupulation density for red swamp crayfish is the concern for production.
Author Response
|
1. Summary |
|
|
|
Thanks very much for the help comments from you and the three reviewers. We have revised the manuscript accordingly and hope that we have properly responded to the comments. Please see our point-to-point responses below. We hope that our revised manuscript may be approved by you. |
||
|
2. Point-by-point response to Comments and Suggestions for Authors Comments 1: [Personality traits (shyness, exploration, and aggression) of juvenile crayfish under varying population densities upon reaching sexual maturity assessed will support the production efficiency. Many factors actually influence the personality traits, which include the population test, body size, gender and so on. However, you have not given the suggestive density for the production. In addition,due to the net factor-sex,, interactive effect between sex and density should be estimated. What is important factor body size should be involved in this test.] |
||
|
Response 1: [Sorry for the confusion. First of all, thank you very much for taking the valuable time to review. We would like to explain that our study focused on investigating the impact of population density on crayfish personality. So we did not consider body size at the time. And experimental crayfish in our study have reached sexual maturity, and all the body size were about 8~10 cm. Factors such as the diversity and variability of external environments, including density, as well as social relationships and resource availability among animals, collectively contribute to shaping animal personalities. We anticipate that this study will provide a more profound understanding of how environmental factors trigger changes in behavioral differences across multiple dimensions. This provides insight into the evolution of animal behavior and personality in different density environments, and how differences in behavioral syndromes at the population level are formed. In the initial stage of model construction, we considered the interaction effect of all influencing factors, but in the subsequent confidence interval analysis, we found that the interaction effect of population density and sex was not significant. Compared with the model with interaction effect, the model without interaction effect had better fit. Therefore, we removed the interaction effect to focus on the main effects. Specific analysis methods in page 6- 7, line 255-273. Thanks again for your careful review and constructive suggestions. We have made corresponding modifications according to your comments and those of the other two reviewers, and we hope that our revised manuscript may be approved by you.
|

Reviewer 2 Report
Comments and Suggestions for Authors
Effect of population density on personality of crayfish (Procambarus clarkii).
Li Su et al. (2024).
The authors investigated the exploratory and aggressive behavior of the crayfish (Procambarus. clarkii) in three population densities.
Main observations:
1. To represent their findings visually, the authors should include photograms of the Shyness, Exploration, and Aggression in the crayfish (Table 2).
2. Authors should best describe Figures 2 and 3, including each X-Y coordinate (units) in the different panels, and they should extend these legends with more detailed information.
3. It is essential to discuss the exploratory behavior in crayfish because it is complex (DOI: 10.1242/Jeb.02020; DOI: 10.1651/S-2687.1, and not only the locomotion activity in crayfish here quantified.
4. Questions:
A. Did you use specialized software to quantify the distance traveled by each crayfish in the new containers with different numbers of animals in the same container? Again, it is convenient to include representative images or photograms.
B. Was each crayfish's personality (shyness and aggressive behavior) analyzed by a researcher or using special software?
Best regards
Author Response
|
1. Summary |
|
|
|||||
|
Thank you for your positive comments on our manuscript. We are also very appreciated for all your suggestions, which have significantly improved our manuscript. Please see our point-to-point responses to your suggestions below.
|
|||||||

Reviewer 3 Report
Comments and Suggestions for Authors
The paper is well written and easy to understand. Just a few comments.
The crayfish were only left to settle for 5 mins - and it takes several hours for animals to settle after handling - not sure if this would affect the animals behavior - not much that can be done about this now
I think the major area for improvement is the discussion. It repeats the results a lot without really saying WHY the authors got the results.
Why does rearing animals in low density make them shy, what happens during the high density raising that makes them aggressive? At present it really reads like - we found this whereas others found that.
In addition, there doesn't appear to be a great difference in density rearing (2, 4, 5) and this shows in the graphs - there is a lot of variation in the points. I wonder why they didn't do more pronounced differences? Either way I think the discussion needs to be reworked at this stage to discuss reasons for behavior rather than a compare and contrast with other studies
Author Response

(The authors gave the same response as above.)
